# Development of an Automatic Alpine Skiing Turn Detection Algorithm Based on a Simple Sensor Setup

**DOI:** 10.3390/s19040902

**Published:** 2019-02-21

**Authors:** Aaron Martínez, Rüdiger Jahnel, Michael Buchecker, Cory Snyder, Richard Brunauer, Thomas Stöggl

**Affiliations:** 1Department of Sport and Exercise Science, University of Salzburg, Schlossallee 49, 5400 Hallein/Rif, Austria; ruediger.jahnel@sbg.ac.at (R.J.); michael.buchecker@sbg.ac.at (M.B.); cory.snyder@sbg.ac.at (C.S.); thomas.stoeggl@sbg.ac.at (T.S.); 2Salzburg Research Forschungsgesellschaft m.b.H., Techno-Z III, Jakob-Haringer-Straße 5, 5020 Salzburg, Austria; richard.brunauer@salzburgresearch.at

**Keywords:** accelerometer, gyroscope, IMU, precision, ski

## Abstract

In order to gain insight into skiing performance, it is necessary to determine the point where each turn begins. Recent developments in sensor technology have made it possible to develop simpler automatic turn detection methodologies, however they are not feasible for regular use. The aim of this study was to develop a sensor set up and an algorithm to precisely detect turns during alpine ski, which is feasible for a daily use. An IMU was attached to the posterior upper cuff of each ski boot. Turn movements were reproduced on a ski-ergometer at different turn durations and slopes. Algorithms were developed to analyze vertical, medio-lateral, anterior-posterior axes, and resultant accelerometer and gyroscope signals. Raw signals, and signals filtered with 3, 6, 9, and 12 Hz cut-offs were used to identify turn switch points. Video recordings were assessed to establish a reference turn-switch and precision (mean bias = 5.2, LoA = 51.4 ms). Precision was adjusted based on reference and the best signals were selected. The *z*-axis and resultant gyroscope signals, filtered at 3Hz are the most precise signals (0.056 and 0.063 s, respectively) to automatically detect turn switches during alpine skiing using this simple system.

## 1. Introduction

The qualitative assessment of alpine skiing is necessary at both the recreational and elite levels. A deeper understanding of the different features related to skiing would allow for improved injury prevention or coaching efficiency. Between elite alpine skiers, the differences in performance are very small, frequently only hundredths of a second [1]. To assess the specific details that lead to those variations in performance, it is necessary to reduce each skiing run into smaller units. Turns are the basic units of alpine skiing; and therefore, it is crucial to precisely determine when each turn begins [2]. The segmentation of skiing runs into turns would allow further enrichment of the data within and between turns (e.g., symmetry, edging angle, acceleration, pressure control…).

There have been different approaches to determine ski turns. Supej et al. [3] were the first to propose a turn switch detection system based on video analysis and with an objective criterion able to detect turns for slalom (SL) and giant slalom (GS). They defined the beginning of the turn as the intersection between the projection of the center of mass (CoM) onto the slope and the arithmetic mean of the skis’ trajectory on the same plane. It was suggested that their methodology would be useful independent of the skiing technique. In 2011, Nakazato et al. [4] developed a methodology based on ground reaction forces generated by force plates integrated between skis bindings and pressure insoles. They defined the moment of edge change (or turn switch) as the minimum value of the vertical ground reaction force determined by the sum of vertical forces from both legs. Fasel et al. [5] defined a turn switch as the point when the lengths of the right ankle joint center to CoM and the left ankle joint to CoM vectors are equal. For the computation of those vectors, they used a setup of seven inertial measurement units (IMU) (right and left shank and thigh, sacrum, sternum and helmet) and a global navigation satellite system (GNSS) attached to the helmet [6,7]. Finally, Yu et al. [8] placed 16 IMUs on an elite skier and used the roll angle (rotation about the anterior-posterior axis) to determine the number of turns performed. The angle relative to the vertical was computed and the interval between two zero angle crossings (shift from leaning left to right, or right to left) was considered a turn. They concluded that the best positions to place IMUs for turn detection were pelvis, shanks and feet.

The development of smaller, long-lasting, wireless technology has made it possible to implement different sensors into the structure of sports equipment, making the collection of data much easier. Furthermore, as the athlete does not feel the sensors, possible alterations in the data collected based on discomfort and/or related technique modifications are avoided. This development has encouraged several attempts to simplify the application of the aforementioned turn detection methods using systems such as instrumented binding (force platforms/bindings) [9,10], video analysis [11], IMUs [5,8,12] and GNSS [5,13]. Although all of these methods provided useful insights in skiing performance, they still present some disadvantages. For example, some require a time consuming set-up, others depend on a very limited capture volume or a small number of turns. Other systems require labor-intensive post processing or a complex set of sensors [7,12]. As a consequence, it is not yet feasible for skiers and coaches to use those systems in a regular basis as a tool to improve performance, provide feedback about motion quality, or reduce injury risk.

The movement of skiing has been often modeled as an inverted pendulum [1,14,15]. The neutral position of the pendulum occurs during straight skiing, when the radial acceleration is zero and consequently, at the point of maximum angular speed [15]. This neutral position corresponds to the unloading phase of the turn and the point of edge changing point [4,8,16], and represents a potential turn detection feature. While similar methods have been used to count turns [8], their precisions have not yet been reported. 

In an attempt to fill the gap between easy to use and reliable methodologies, we aimed to develop a sensor set-up using IMUs and the necessary algorithms to detect turns precisely. The objective of the system is to be simple, comfortable and feasible to use on a daily basis. We hypothesized that the local extrema of angular velocity signal would correspond with the edge changing and thus, with the beginning of the turn.

## 2. Materials and Methods

To identify the most precise automatic methodology to detect turns in alpine skiing, an experiment was designed where an IMU was attached to the upper posterior cuff of each ski boot. The protocol was performed on a ski-ergometer based on a commercially available simulator (Proski Simulator—Trgovina in storitve, Rače, Slovenia, see Figure 1). The basic unit was reinforced to resist the high mechanical stress and assembled with a custom-manufactured slide board. The slide board was modified with an alpine ski binding for a more specific ski simulation. The bindings were mounted on a plate which served to lift the respective “inside heel” to enable ski specific knee and hip angulation [17]. Compared with the commercial slide board “Proski simulator”, the custom-manufactured ergometer was attached with clamping rolls to ensure that the board cannot derail. The different test conditions reproduced the turn movement of a skier at different turn durations and slope conditions. Acceleration and gyroscope signals in three axes plus the resultant signals were filtered at different frequencies and analyzed in order to develop automatic algorithms. Detected turns were compared to gold standard expert video analysis to determine the most precise methodology with respect to signal, axis and filter.

### 2.1. Participant

The participant was a 31 year-old male expert skier (height: 174 cm; mass: 69 kg). Before the measurement, he was informed in detail about the testing procedures, as well as possible benefits and risks of the investigation prior signing the consent form approved by the local Ethics Committee (EK-GZ: 11/2018). The experiment was conducted in accordance with the Declaration of Helsinki.

### 2.2. Instruments

An IMU (LSM6DS3, 2.5 × 3 × 0.83 mm, ±8 g and ±500 dps full scale, STMicroelectronics, Amsterdam, Netherlands) was placed in the back of the upper cuff of each boot (Hawx 130, Atomic, Altenmarkt, Austria). The X axis of the IMU was aligned with the vertical axis of the boot pointing superiorly, the Y with the lateral axis pointing to the left, and the Z with the roll axis pointing posteriorly. 3D Angular velocity and acceleration signals were recorded at 833 Hz, after an analog low-pass filter and a digital low-pass filter after the ADC, the raw data extracted from the sensor had a rate of 64 Hz. The sensor data was transmitted via Bluetooth and collected by an in-house smartphone application (SkiSense App, Salzburg Research, Salzburg, Austria). 

For precise determination of the true turn switches (gold standard) a camera (Basler, Ahrensburg, Germany) sampling at 50 Hz, was placed perpendicular to the frontal plane of the ski-ergometer. The IMU data was resampled to 50 Hz using a shape-preserving spline interpolation algorithm in order to match the sampling rate of the camera. 

### 2.3. Experimental Situations

Three different slope conditions were recreated using different ski-ergometer positions. In the first position the ski-ergometer was placed flat on the floor of the laboratory. In the second position, the ergometer was leaned forward at an inclination of 6.3°. In the third position, the ergometer was leaned to the left at an inclination of 5.7°. Based on the average duration of GS and SL turns [18], two different turn durations (1.45 and 0.90 s) were performed for each slope condition and controlled with a metronome that the skier had to match. At least 15 turns were recorded for every condition.

### 2.4. Reference Turn Switch Value

The instant of turn switch for each turn was assessed by three experts (an Austrian skiing instructor, a world cup skier and a skiing consultant) using the video recordings. One way repeated measures ANOVA (SPSS Inc.; Version 25.0, Chicago, IL, USA) with Bonferroni post-hoc tests were conducted to test for differences between raters. The maximum level of precision and evaluation of bias were calculated using the limits of agreement (LoA) as proposed by Bland and Altman [19] (Excel 2016, Microsoft, Redmond, WA, USA). The 95% limits of agreement were estimated by mean difference and ±1.96 standard deviation of the differences. The average turn switch between the three raters for each turn was used as the reference turn switch point for further analysis.

### 2.5. Data Analysis

The eight signals (X, Y, Z axes and resultant of the gyroscope and the accelerometer data) were visually inspected. The signals were synchronized with the reference turn-switch points identified in the video analysis. Features and patterns were identified from the synchronized data to develop turn detection algorithms. The development process consisted of various steps and was individual to each signal, or group of signals with similar characteristics. First, semi-automatic algorithms were developed in order to find the landmarks for every signal and filter cut-off option. The semi-automatic algorithms consisted of a strong low pass filter (0.5 or 1 Hz) to determine the turn switch window. Afterwards, each signal was filtered using a fourth-order zero-lag low-pass Butterworth filter with cut-off frequencies of 3, 6, 9 and 12 Hz. The implementation of different filter options increased the number of variables to 120: two sensors (accelerometer and gyroscope), four axes (x, y, z and resultant), three signals (right boot, left boot and average) and five filtering options (3, 6, 9, 12 and no filter). Local maxima, corresponding to the turn switch point, were identified within the windows defined by the strong low-pass filter using the *findpeaks* function provided by MATLAB (R2017B, MathWorks, Natick, MA, USA). Thresholds for prominence, separation, and distance were adapted for each signal. The semi-automatic algorithms developed for the signals with best precision were further developed into automatic algorithms (see Section 3.1 Algorithm development for a detailed description of the suggested algorithms). All the algorithm development and assessment was performed with MATLAB.

The results for the three different slope conditions and turn lengths were pooled to reproduce true skiing conditions where changes in slope gradient, inclination and turn duration occur. This procedure increased the potential robustness of the turn detection methodology by replicating in-field conditions during the development process. Mean bias between each variable and the reference value, and 95% LoA [19] in ms were calculated. To check if the data was normally distributed, Saphiro-Wilks tests were performed.

The precision of a measurement system cannot be higher than its reference system or gold standard. To express the precision of the measurement in terms of the reference precision, the measured LoA were adjusted to the reference system precision. The LoA of the reference were defined as the highest upper limit and the lowest lower limit between the three raters. The measured precision was adjusted as follows: if the absolute values of the upper and lower measured LoA were higher than the absolute values of the reference LoA, the upper and lower reference LoA were subtracted from the upper and lower measured LoA (see Figure 2a). If the upper or lower measured LoA was within the reference LoA range, the measured value was replaced by the reference value (see Figure 2b,c). If both measured LoA were outside and greater than the upper or lower reference LoA, the measurement methodology was discarded (see Figure 2d). This ensured that the precision of the measurement was never better than the reference precision. Henceforth, the adjusted precision is reported as the sum of the adjusted measured LoA absolute values.

The selection process to determine the best automatic turn-switch detection system consisted of three steps:1)The variables where the bias was outside the reference LoA were excluded.2)The variables with an adjusted precision higher than 20 ms were excluded.3)The lowest adjusted precision was select as the best methodology.

## 3. Results

The results of the bias and 95% LoA between raters are reported in Table 1. The one-way repeated measures ANOVA showed a difference between raters. A bias of 3.2 (*p* = 0.015) and 5.2 ms (*p* < 0.001) was found. The maximum precision of our reference system was set as the difference between the highest upper limit, in this case 30.0 ms and the lowest lower limit, –21.4 ms. Consequently, the maximum precision of the reference system is 51.4 ms.

The steps of the method selection process along with its results are presented in Table 2. The data was not normally distributed, therefore it was also analyzed using the 2.5th and 97.5th percentiles to ensure the 95% confidence interval (CI). The variables with a bias outside the reference LoA are marked with a strikethrough and removed from further analysis (selection process step 1). The variables highlighted in the last three columns have an adjusted precision value lower than 1 (selection process step 2). The resultant and the *z*-axis (Figure 3) of the gyroscope consistently showed adjusted precision less than 20 ms for all filter and signal options (with the exception of the *z*-axis 12 Hz right boot and the resultant of the left boot with no filter). While the resultant acceleration in the right boot showed adjusted precision values lower than 1, both the left and average showed greater values. Consequently, the acceleration signals were not chosen as a reliable method to assess turn switches.

Within the gyroscope data, the filter option that showed the best measured precision was the 3 Hz filter. The gyroscope data in the *z*-axis and the resultant had the best measured precision. The best values were average *z*-axis (6.6 ms), left boot *z*-axis (6.2 ms) and average resultant (11.6 ms).

### 3.1. Algorithm Development

The proposed algorithms for the best signals (e.g. gyroscope *z*-axis and gyroscope resultant, see Figure 3) are shown schematically in Figure 4. 

The gyroscope data during turns were examined for distinguishing characteristics that were consistent within the signals of all slope conditions. An algorithm was devised that precisely detected only turn switch points. The algorithms are slightly different for each signal, both are described as follows.

Algorithm for gyroscope z-axis
*Step 1*:If the desired signal is the average between both boots, calculate the mean of the raw data from both boots. If using the signal from one boot, proceed to step 2. *Step 2*:Filter the signal using a fourth-order zero-lag low-pass Butterworth filter with cut-off frequency of 0.5 Hz. *Step 3*:Find the local extremes in the filtered signal. *Step 4*:Filter the signal from step 1 using a fourth-order zero-lag low-pass Butterworth filter with the desired cut-off (3 Hz recommended).*Step 5*:Find the local maxima in the signal from step 4 between the local minima from step 3.*Step 6*:Find the local minima values in the signal from step 4 between the local maxima from step 3.*Step 7*:Organize values from steps 5 and 6 in ascending order to represent proper turn switch points.

Algorithm for gyroscope resultant
*Step 1*:Calculate the resultant of the 3 axes (x, y and z) for one or both boots (depending in the signal to analyze is one boot or the average between both).*Step 2*:If the desired signal is the average between both boots, calculate the mean of the resultant from both boots. If using the signal from one boot, proceed to step 3. *Step 3*:Filter the signal using a fourth-order zero-lag low-pass Butterworth filter with cut-off frequency of 1 Hz. *Step 4*:Find local minima in the filtered signal.*Step 5*:Filter the signal from step 2 using a fourth-order zero-lag low-pass Butterworth filter with the desired cut-off (3 Hz recommended).*Step 6*:Find local maxima in the signal from step 5 between the local minima from step 4.*Step 7*:The local maxima values represent the turn switch points. 

It should be noted that there are two different filtering processes. The first process goal (e.g. Step 3 in the *z*-axis algorithm and Step 4 in the resultant algorithm) is to determine the individual windows where to look for the turn switch points. The second process (e.g. steps 5 and 6 in the *z*-axis algorithm and 6 in the resultant algorithm) was developed to find the actual turn switch points.

## 4. Discussion

The purpose of this study was to develop a sensor set up and algorithm that precisely detects turn switches in alpine skiing and is feasible for daily use. The results indicate that the proposed system is able to determine the turn switch point with a precision of ±0.03 s using one or two IMUs mounted to the cuff of the boot. As hypothesized, the roll axis of the gyroscope was the signal from which the turn switch could be most precisely detected. The roll axis in the gyroscope was not the only signal that met our inclusion criteria. The resultant signal from the gyroscope (for a single boot or the averaged signal) and the resultant acceleration for the right boot also presented high adjusted precision. It is important to note that the data collection was performed in a highly controlled lab environment, where the movement was limited to one plane. With this particular arrangement, it is evident that the resultant values were predominantly determined by the axis in the plane where the movement was allowed. In the field, where motion is not restricted to a single plane, the resultant may be more affected than other axes. The freedom of movement, the vibrations associated with skiing, and the conditions of snow and slope could introduce noise, affecting the resultant to a greater extent than the *z*-axis signal. Due of the pendulum like movement about the *z*-axis that occurs during skiing [1,14,15], the main characteristics of the signal would be less affected, and the peak angular velocity would still represent the turn switch. Moreover, the resultant of the acceleration was not consistent enough to be used as a reliable indicator. Surprisingly, in such a controlled set up, only the results for the acceleration in the right boot met the inclusion criteria while the values for the left boot and average were drastically less precise (see Table 2). Factors such as attachment or placement of the IMU were discarded as a reason for those results (the IMU was firmly attached and by using the resultant the placement should not be relevant); we concluded that differences between right and left leg movement patterns defined this difference. That this effect was only observed in the accelerometer signal is likely due to the specific nature of the measurement systems. Consequently, we would recommend the use of the *z*-axis gyroscope signal for turn switch detection. 

The results showed similar precision levels between a single boot signal and the average between boots (gyroscope *z*-axis). In consequence, we suggest that a single IMU placed in the back of the upper cuff of one ski boot is the simplest method for turn switch detection. Nonetheless, depending on the desired future measures, it might be relevant to use IMUs on both boots. This could potentially measure and compare performance metrics such as edge angle, symmetry, G-forces, etc.

As reported in the results, the data was not normally distributed. However, the results displayed by the 2.5th to 97.5th percentile presented values comparable to the 95% LoA. The two best signals and cut-off filter were the same. Due to the discrete nature of the data and the type of analysis, the ranges determined by 95% CI based on the percentiles were shifted. Bland and Altman [19] is the main tool to analyze agreement between systems and determine if they are interchangeable. The Bland and Altman analysis showed the limitations of the system (e.g. the differences between raters and a 0.05 s range), while centering the range on the mean. Consequently, we decided to report the results of the Bland and Altman methodology.

A limitation of this study was that the results were based on data from one participant. However, the aim of the study was to develop automatic algorithms and to select the most precise signal and filter cut-off options in a well-standardized laboratory situation (e.g. ski ergometer). Based on these results, more research is needed, where more participants are involved in an ecologically valid environment to enhance the external validity of this concept. While on-snow skiing, the plane of movement and the skiing technique are not limited, as it is the case on the ski-ergometer. In further field based studies, a broad variety of external conditions (e.g. snow condition, slope gradients, ski geometries, etc.), skiing styles, and skill levels will introduce adequate and relatable variability to the signals allowing further development and scientific proof of the algorithm. 

A pilot test was performed on snow to have the first insight to the system’s performance during in-field alpine skiing. An expert skier performed a complete downhill run (of carved turns) and the turns counted by the system were compared to the turns counted using a wearable kinematic global navigation satellite system (AT-H-02, AOBA Technologia LLC, Sendai, Japan) measuring at 10 Hz. With both systems the same number of turns (N = 28) were counted and the characteristics for the selected signals (e.g. *z*-axis of the gyroscope) were very similar. However, further research needs to be done in order to assess the validity and robustness of the system during on-snow skiing with different skiers, slope and snow conditions and skiing styles.

To our knowledge, there was only one previous study reporting precision of turn detection methodologies. Fasel et al. [5] assessed the accuracy and precision of two different turn detection methods. However, there are few aspects to take into consideration when comparing their results to the present study. The methodologies to calculate the reference values are different. Fasel et al. used a concept proposed by Supej et al. [3]. They based their calculations on the estimation of the CoM and ski trajectories using GNSS and IMUs, and did not report values against video recordings. Conversely, the reference values in the current study are based on expert analysis of video recordings. By placing the camera perpendicular to the plane of motion, raters had the best possible perspective to evaluate actual points of turn switch. However, a small but significant bias was observed between raters. Consequently, it is logical to interpret that measurements on the field would have less inter-rater precision, highlighting the importance of assessing the precision of the reference system. To address this issue, we averaged the reported turn switch point of three raters and performed agreement analysis between raters, showing a maximum LoA difference (95% CI), or a maximum turn detection precision of 0.05 s.

The calculations used to assess the system’s precision in this study are different than the calculations used by Fasel et al. [5]. We based our maximum inter-rater (reference) precision and methodology evaluation on LoA, ensuring a 95% probability that every detected turn would be within our range. Conversely, the accuracy and precision reported by previous studies, were based on average differences and standard deviation respectively. Those calculations do not guarantee that all turns will fall into the precision range. Following the methods from Fasel et al. [5], our values for accuracy and precision between the selected signal (e.g. gyroscope average 3 Hz) and the mean reference value (mean between the three raters) were 0.3 ms and 14 ms, respectively. Using the same calculations, our inter-rater reference accuracy and precision improved to 16 ms and 15 ms respectively. Although the precision of a measurement methodology cannot be better than the gold standard, both studies reported values (based on average difference and standard deviation) smaller than the sampling rate. Therefore, these measurements do not reflect the reality of the system. As a solution, the present study proposes a method that ensures that the measured turn switches will be within the precision range with a 95% confidence, while accounting for the differences in exact turn points between raters.

## 5. Conclusions

In conclusion, the algorithm and sensor set-up proposed in this study is a precise and simple system to automatically detect turn switches during alpine skiing. Moreover, this methodology is non-obtrusive, wireless, and could be implemented in a smartphone application, providing semi-instantaneous feedback. Further research is necessary to implement this algorithm in the field and evaluate its validity with respect to different skier abilities, slope conditions and turn styles. It also represents a first step in order to develop new measurements to better understand skiing technique and performance. 

## Figures and Tables

**Figure 1 sensors-19-00902-f001:**
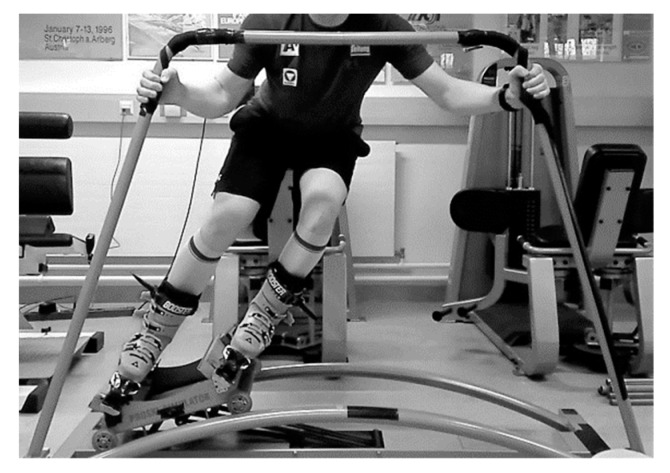
Ergometer.

**Figure 2 sensors-19-00902-f002:**
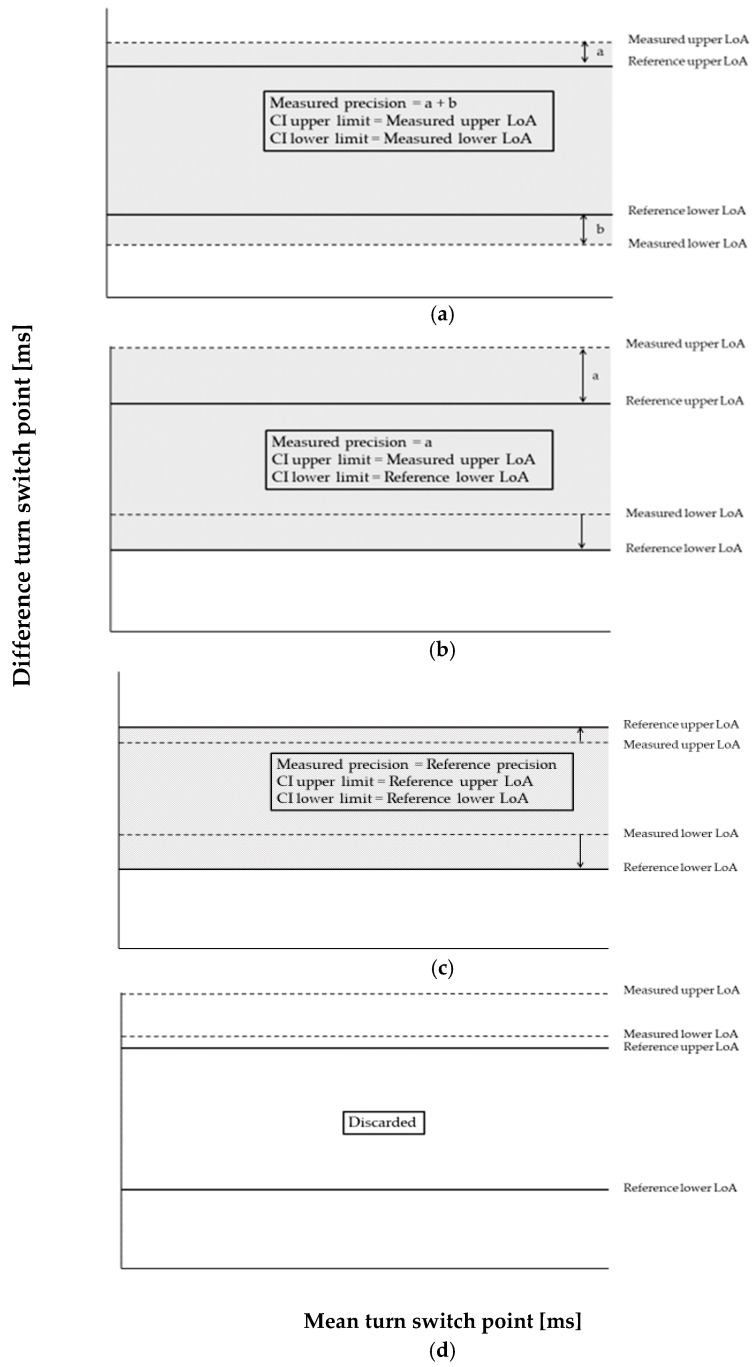
Different measured LoA adjustment possibilities. (**a**) Upper and lower measured LoA are outside the reference LoA range. In this case the upper and lower reference LoA are subtracted from the upper and lower measured LoA. The CI (grey area) is the range between the measured LoA. (**b**) One measured LoA is inside and one outside of the reference LoA range. In this case the measured LoA inside the range is replaced with its correspondent reference LoA, the outside measured LoA is adjusted as case a. The CI is the range between one measured LoA and the replaced LoA. (**c**) Both measured LoA are inside the reference LoA. Both values are replaced by the reference LoA. The CI is the range between the replaced LoA. (**d**) Both measured LoA are outside the reference LoA range and and greater than the upper or lower reference LoA. The measurement methodology was discarded. CI, confidence interval; LoA, limits of agreement.

**Figure 3 sensors-19-00902-f003:**
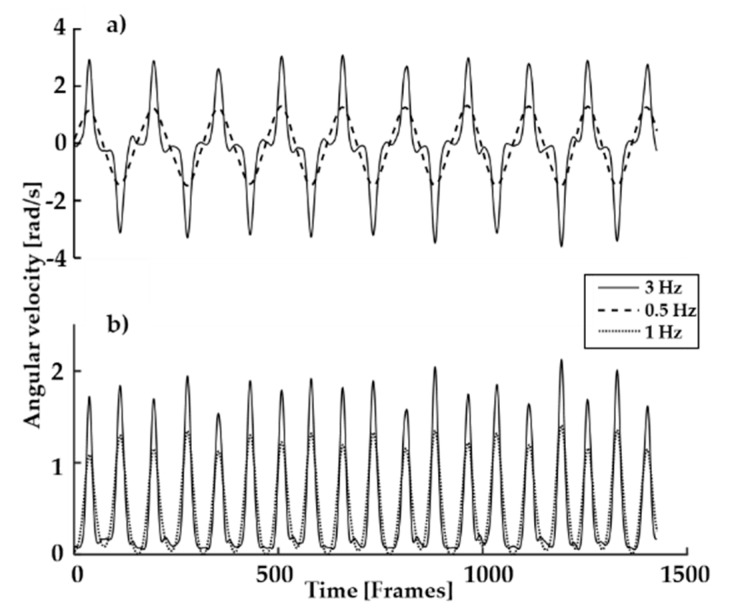
Angular velocity (gyroscope) signal averaged from both boots and filtered at 3 and 1 Hz. Figure **a**) shows the signal from the *z*-axis and **b**) shows the resultant.

**Figure 4 sensors-19-00902-f004:**
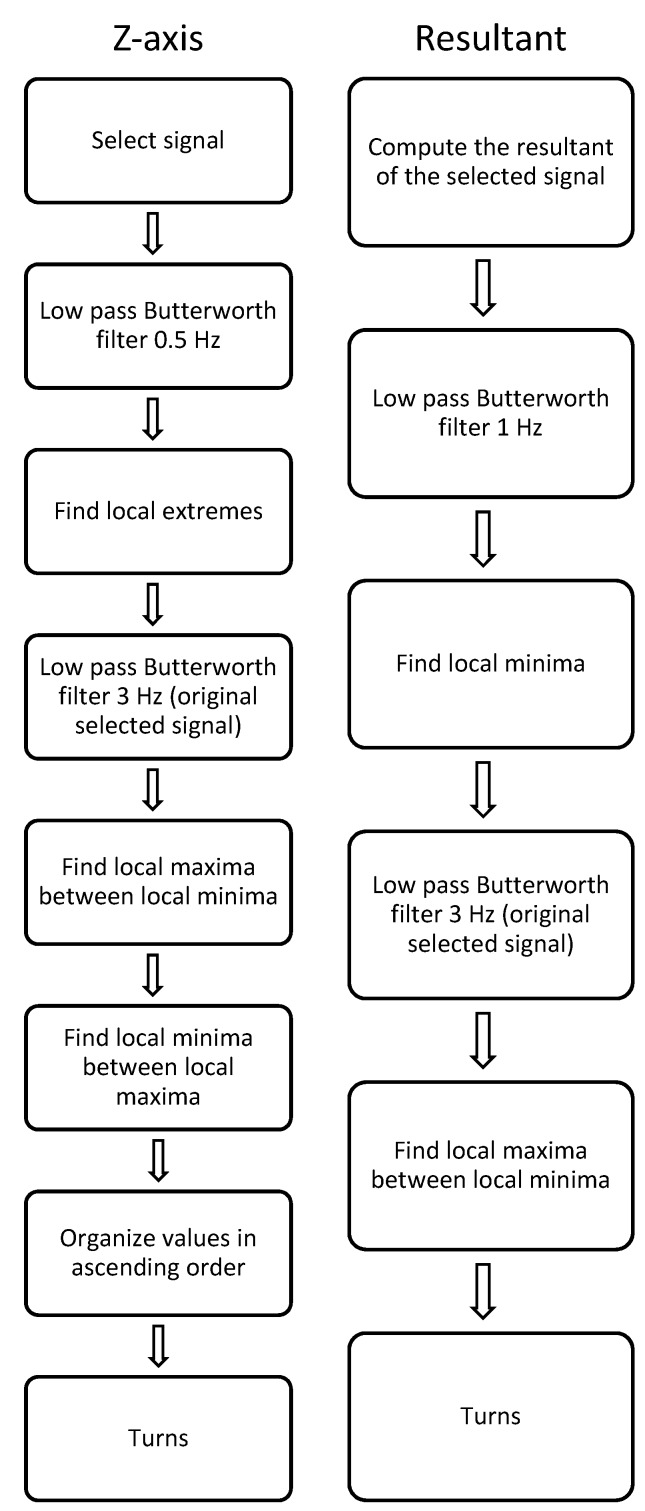
Schematic representation of the algorithms of turn detection in the *z*-axis and resultant of the gyroscope signal.

**Table 1 sensors-19-00902-t001:** Inter-raters bias, limits of agreement and maximum precision (difference between limits).

Rater	Bias (ms)	Upper Limit (ms)	Lower Limit (ms)	Maximum Precision (ms)
1 vs. 2	2	23.4	−19.6	43.0
1 vs. 3	3.2 *	28.0	**−21.4**	49.4
2 vs. 3	5.2 *	**30.0**	−19.8	49.8

* significantly different.

**Table 2 sensors-19-00902-t002:** Bias, limits of agreement and measured precision. The selection process is as follows, the variables with a bias outside the reference LoA are marked with a strikethrough and removed from further analysis (step 1). The variables highlighted in the measured precision are the ones with a value lower than one (step 2).

			Bias (ms)				Lower Limit (ms)				Upper Limit (ms)				Adjusted Precision (ms)			
	Sensor	Filter	Resultant	Z	Y	X	Resultant	Z	Y	X	Resultant	Z	Y	X	Resultant	Z	Y	X
**Average**	**GYRO**	0	−2.8	2.4	9.4	−7.6	−35.2	−29.8	−53.6	−88.2	29.4	34.4	72.4	73.2	**13.6**	**12.8**	74.6	110
		3	−5.4	0.2	5.8	−10.6	−33.0	−28.0	−34.0	−63.4	22.2	28.6	45.8	42.4	**11.6**	**6.6**	28.2	54.2
		6	−3.2	2.6	8.2	−9.4	−34.0	−29.2	−36.4	−80.0	27.6	34.4	52.8	61.2	**12.6**	**12.0**	37.6	89.6
		9	−3.2	2.6	12.2	−9.2	−34.8	−29.8	−48.2	−83.4	28.4	35.0	72.4	65.2	**13.2**	**13.2**	69.0	97.0
		12	−3.4	2.2	12.2	−8.8	−36.0	−31.2	−49.4	−83.6	29.0	35.6	73.8	65.8	**14.4**	**15.4**	71.6	97.8
	**ACC**	0	−10.8	−29.8	−12.6	−111	−79.6		−42.2		58.0		16.8		86.0		20.6	
		3	−15.0	−5.0	−13.6	−57.4	−79.4	−388	−40.0		49.6	378	12.6		77.4	715	**18.4**	
		6	−9.8	−4.2	−13.8	−62.8	−32.0	−371	−40.4		12.4	362	12.6		**10.6**	681	**19.0**	
		9	−11.0	−12.4	−13.8	−65.2	−12.4	−362	−41.8		57.6	337	14.2		85.6	648	20.2	
		12	−10.4	−19.8	−12.8	−67.0	−79.8	−350	−42.4		59.0	310	16.6		87.2	609	20.8	
**Right**	**GYRO**	0	−5.0	−0.8	−0.2	7.6	−39.4	−35.6	−67.6	−163	29.4	34.0	67.0	178	**18.0**	**18.2**	83.2	290
		3	−5.2	0.4	4.8	4.0	−36.8	−32.0	−40.6	−91.2	28.2	32.8	50.2	99.4	**15.2**	**13.2**	39.4	139
		6	−4.4	2.2	7.0	17.4	−37.0	−32.4	−45.2	−101	28.2	36.6	59.2	136	**15.6**	**17.6**	53.0	185
		9	−4.0	2.4	8.2	16.2	−38.2	−32.4	−49.8	−104	30.2	37.2	66.0	136	**16.8**	**18.0**	64.2	189
		12	−4.2	2.6	8.4	19.4	−37.6	−33.6	−48.6	−106	29.4	38.8	65.6	144	**16.2**	20.8	62.6	199
	**ACC**	0	−9.4	−54.6	−14.4	−158	−39.6		−47.6		21.0		19.0		**18.2**		26.2	
		3	−9.0	−60.2	−13.0	−73.2	−71.0		−40.2		53.0		14.4		72.4		**18.8**	
		6	−9.8	−66.2	−13.8	−104	−35.4		−42.2		15.8		14.4		**14.0**		20.6	
		9	−7.6	−50.8	−14.2	−116	−36.6		−43.4		21.2		15.0		**15.0**		22.0	
		12	−7.2	−60.4	−14.8	−140	−38.8		−45.8		24.2		16.4		**17.4**		24.4	
**Left**	**GYRO**	0	−3.4	1.2	4.8	58.8	−39.4	−34.2	−74.0		32.8	36.4	83.6		20.8	**19.2**	106	
		3	−5.2	0.4	12.0	70.2	−34.4	−27.8	−48.0		23.8	28.6	72.0		**12.8**	**6.2**	68.4	
		6	−3.0	3.2	13.4	76.8	−34.4	−28.6	−53.4		28.6	34.8	80.0		**13.0**	**11.8**	81.8	
		9	−2.6	3.6	13.4	72.0	−36.0	−29.4	−55.8		30.8	36.4	82.6		**15.2**	**14.2**	86.8	
		12	−2.6	3.6	14.2	64.8	−38.2	−30.8	−55.4		33.0	38.2	84.0		**19.6**	**17.4**	88.0	
	**ACC**	0	−11.0	−52.0	−10.4	−125	−81.2		−50.2		59.2		29.6		88.8		28.8	
		3	−14.4	−60.2	−16.8	−39.8	−82.6		−86.8		54.0		53.4		85.2		88.6	
		6	−13.2	−61.6	−13.6	−46.0	−81.0		−41.2		54.8		14.0		84.2		**19.8**	
		9	−11.4	−54.2	−12.6	−77.4	−80.6		−42.0		57.6		16.8		86.6		20.4	
		12	−11.0	−49.6	−12.6	−87.2	−80.8		−44.2		59.0		18.8		88.2		22.6	

GYRO, gyroscope signal; ACC, accelerometer signal.

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
