# Peer review of "Development of an Automatic Alpine Skiing Turn Detection Algorithm Based on a Simple Sensor Setup"

_sensors, 2019, doi:10.3390/s19040902_

Round 1

Reviewer 1 Report

Thank you for changing the papers. I'm unfortunately still not convinced by your paper and modifications and have to insist on the following points. They should not be new to you and were already raised in the first review.

Major remarks:

Units: I do not agree with your answer that frames as units simplifies reading and interpretation. You have the same amount of commas and decimals. Frames as units makes interpretation more complicated as one has always to convert in the head to seconds. I would stick with the guidelines and report everything in SI units, i.e. seconds. If you don't want to change, then repeat the frame rate at least in each table description.

LoA and statistics: in the "answer to the reviewer" document you state that your data is not normally distributed and that you checked for it. Thus, you can no longer use standard deviations. What I suggest doing here is that you add something to the text along the same lines as you reported in the "answer to reviewer" document: that data was not normally distributed but that you checked that 2.5th and 97.5th percentiles where close to the 95% LoA and that you therefore decided to keep the numbers you already have. You could then put the percentile values and comparison in the appendix.

Method presentation: I also agree with Reviewer 3 and suggest strongly to move the algorithm to the methods section. It must be known to the reader what algorithm you used for every axis and sensing dimension (acceleration, angular velocity), prior to presenting which choice of filtering and axes provides the best results. So did you use that exact same algorithm for everything? And why did you not test on zero crossings, minima, etc.?

Line 76: "replace" maxima by "extrema"

Line 107: sampling and low pass filtering. Your text does not match the data sheet for the LSM6DS3 (https://www.st.com/content/ccc/resource/technical/document/application_note/12/98/b4/44/a5/bf/4e/c5/DM00157511.pdf/files/DM00157511.pdf/jcr:content/translations/en.DM00157511.pdf). There is no sampling frequency of 64Hz. Moreover, contrary to what has been stated in your "answer to reviewers" document, according to section 3.8 of the same document, you *MUST* have a low pass filter during A/D conversion. Please comment.

Line 118: write that inclination is to the left

Line 247: add the response you gave me in the text: if one has to choose, then the norm might be more robust than only the z-axis.

Lines 275 - 277: now that the text is clear I can follow your argumentation and see that you are unfortunately not entirely correct. Assuming normal distribution of the error, having mean error (accuracy) and standard deviation (precision), one can always compute the LoA, you even provide the definition how (Line 127). So those two concepts are exactly the same. Also be careful what LoA means: it is a statistical measure, where 95% of the data are supposed to fall within this range. But if you record 100 turns, then 5 of them have worse precision.

Lines 279, 280: Convert units to seconds

Spelling:

Abstract: Line 24: there seems to be some zero missing

Line 107: space missing between "angular velocity and" and "acceleration". Angular velocity should be lower case.

Line 142: Comma missing between MA and USA

Line 168: what are the units?

Line 239: Replace "precision" by "precise"

Author Response

Reviewer #1:

Thank you for changing the papers. I'm unfortunately still not convinced by your paper and modifications and have to insist on the following points. They should not be new to you and were already raised in the first review.

A: Dear reviewer, thank you again for your valuable comments and recommendations and for the time invested on helping to improve the quality of the manuscript. We hope to have made the appropriate modifications. Please find the changes related to your comments in “blue”.

Major remarks:

Units: I do not agree with your answer that frames as units simplifies reading and interpretation. You have the same amount of commas and decimals. Frames as units makes interpretation more complicated as one has always to convert in the head to seconds. I would stick with the guidelines and report everything in SI units, i.e. seconds. If you don't want to change, then repeat the frame rate at least in each table description.

A: We changed the units. After discussing your comment, we agreed that frames might bring confusion to the reader and we decided that ms would be the unit that would make the paper easiest to read. Consequently, most of the measures and all the tables are in ms, and some of the main results are expressed in s.

LoA and statistics: in the "answer to the reviewer" document you state that your data is not normally distributed and that you checked for it. Thus, you can no longer use standard deviations. What I suggest doing here is that you add something to the text along the same lines as you reported in the "answer to reviewer" document: that data was not normally distributed but that you checked that 2.5th and 97.5th percentiles where close to the 95% LoA and that you therefore decided to keep the numbers you already have. You could then put the percentile values and comparison in the appendix.

A: We added the suggested information. Please see L180-L182 & L264-L271

Method presentation: I also agree with Reviewer 3 and suggest strongly to move the algorithm to the methods section. It must be known to the reader what algorithm you used for every axis and sensing dimension (acceleration, angular velocity), prior to presenting which choice of filtering and axes provides the best results. So did you use that exact same algorithm for everything? And why did you not test on zero crossings, minima, etc.?

A: We discussed again about that point and we still believe that the final algorithm is the main result of the study. However, we can also see the point raised by Reviewers #1, 2 & 3  that the “development of the algorithm” should be part of the methods. Therefore, we modified the section 2.5 Data analysis in order to make the process very clear to the reader. In the first step, semi-automatic algorithms were developed and landmarks were found. With those results, precision was measured and the best signals were selected. The algorithms for those signals were further developed and made automatic. We selected the landmarks that corresponded to the turn switch points detected by the raters; consequently, we developed algorithms to find those particular landmarks, this is the reason why zero crossing & minima are not tested. See L133-L150.

 Line 76: "replace" maxima by "extrema"

A: Changed as suggested. See L76

Line 107: sampling and low pass filtering. Your text does not match the data sheet for the LSM6DS3 (https://www.st.com/content/ccc/resource/technical/document/application_note/12/98/b4/44/a5/bf/4e/c5/DM00157511.pdf/files/DM00157511.pdf/jcr:content/translations/en.DM00157511.pdf). There is no sampling frequency of 64Hz. Moreover, contrary to what has been stated in your "answer to reviewers" document, according to section 3.8 of the same document, you *MUST* have a low pass filter during A/D conversion. Please comment.

A: Thank you for highlighting this point. The chip uses an analog low-pass filter before and a digital low-pass filter after the ADC for anti-aliasing. Because of the sampling rate of 833 Hz, the internal cutoff frequencies for the low pass filters are much higher than the data rate of the devices. We had more sensors sending data and the transmission band-with was limited making 64 Hz the best option. Therefore, the raw data extracted from the sensor after the internal filters was at 64 Hz. See L107-109

Line 118: write that inclination is to the left

A: added as suggested. See L119

Line 247: add the response you gave me in the text: if one has to choose, then the norm might be more robust than only the z-axis.

A: We added our recommendation between the 2 options, in this case we believe that the z-axis would be more robust than the norm. The explanation can be found in L246-L258.

Lines 275 - 277: now that the text is clear I can follow your argumentation and see that you are unfortunately not entirely correct. Assuming normal distribution of the error, having mean error (accuracy) and standard deviation (precision), one can always compute the LoA, you even provide the definition how (Line 127). So those two concepts are exactly the same. Also be careful what LoA means: it is a statistical measure, where 95% of the data are supposed to fall within this range. But if you record 100 turns, then 5 of them have worse precision.

A: We agree with your comment, the computation from mean error and standard deviation to LoA is possible. However, we do not propose a different nomenclature, our proposed method is based on adjusting the precision to adapt the values to the maximum precision determined by the raters. In other words, we modify how to report the values in order to allow them to be adjusted to the values of maximum precision based on the raters agreement. See L156-L167.

Lines 279, 280: Convert units to seconds

A: as we mentioned before, we changed most of the values to ms in order to have a unit not depending on the sampling rate and easy to interpret. So we decided to leave those particular values in ms.

Spelling:

Abstract: Line 24: there seems to be some zero missing

A: Thank you! Changed as suggested. See L24

Line 107: space missing between "angular velocity and" and "acceleration". Angular velocity should be lower case.

A: Changed as suggested. See L107

Line 142: Comma missing between MA and USA

A: Changed as suggested. See L146

Line 168: what are the units?

A: Thank you. The units now are ms. Changed as suggested. See L176

Line 239: Replace "precision" by "precise"

A: Changed as suggested, thank you. See L253

Reviewer 2 Report

The  authors have addressed most of the detailed comments appropriately.  However, there are still some main critical issue that were not properly  tackled.

1)      Considering that is a laboratory  test, the type of algorithm proposed (finding local maxima and minima),  and the parameter investigated (filter cut-offs), in order to give  sufficient rights of novelty and robustness  of the method proposed, more athletes must be acquired or results  coming from in-field test must be provided. Considering the type of  acquisition, acquiring more athletes in laboratory should not be so  cumbersome and would give more robustness to the results.  The authors hypothesized that little differences among athletes in this  type of test would be found, but this must be verified and not only  hypothesized.

2)      The description of the algorithm  cannot be considered a result. Of course, you find an algorithm that  fits the aim of the study, and you have developed the algorithm  investigating different variables or data  processing. When a new algorithm is proposed this is always the case.  However, the results are the performance of the algorithm when it is  tested (in laboratory or in field situation) and not the algorithm  itself. The description of the algorithm must be shifted  in materials and method.

Detailed comment

Pag3 line 109-110, please clarify why this specific values of slopes were chosen (6.3°,5.7°)

A:  The inclination values were not based on literature. We wanted to check  the performance of the algorithm in different conditions so we used the  equipment available for us.

R:  You must give an explanation that is not “we used the condition  available in the equipment”, at least you must compare this value with  literature or infield data, and justify why you considered  this values sufficient to test the performance of the algorithm.

Author Response

Reviewer #2:

The  authors have addressed most of the detailed comments appropriately.  However, there are still some main critical issue that were not properly  tackled.

A: Dear reviewer, thank you for your effort in helping us to improve the quality of the manuscript. We hope this version and our explanations are clearer. The changes related to your comments are marked in “green”.

1)      Considering that is a laboratory  test, the type of algorithm proposed (finding local maxima and minima),  and the parameter investigated (filter cut-offs), in order to give  sufficient rights of novelty and robustness  of the method proposed, more athletes must be acquired or results  coming from in-field test must be provided. Considering the type of  acquisition, acquiring more athletes in laboratory should not be so  cumbersome and would give more robustness to the results.  The authors hypothesized that little differences among athletes in this  type of test would be found, but this must be verified and not only  hypothesized.

A: The methodology proposed was with the aim of developing a first step towards an algorithm that works in the field. We wanted to determine which signal and filter cut-off options were more precise, and we believe the current methodology allow for those conclusions. The acquisition of more athletes in a laboratory set up, would definitely bring more variability to the data. However, this variability would be restricted by the plane of movement and the bindings of the glider. The variability that we would find on piste during on-snow skiing would be most likely based on different terrain conditions, techniques, vibration… and this is not measurable with the glider. We are planning the second part of this algorithm development which consists on evaluating and further developing the algorithm collecting data from different skiers in the field. This type of data collection will bring higher within and between subject variability to the signals and will allow us to enhance the algorithm with real skiing data. See L272-L280

2)      The description of the algorithm cannot be considered a result. Of course, you find an algorithm that fits the aim of the study, and you have developed the algorithm investigating different variables or data processing. When a new algorithm is proposed this is always the case.  However, the results are the performance of the algorithm when it is tested (in laboratory or in field situation) and not the algorithm itself. The description of the algorithm must be shifted in materials and method.

A: As answered to reviewer #1, “we discussed that point and we strongly believe that this is the main result of the study. However, we modified the section 2.5 Data analysis in order to make the process very clear to the reader. In the first step semi-automatic algorithms were developed and landmarks were found. With those results, precision was measured and the best signals were selected. The algorithms for those signals were further developed and made automatic. We selected the landmarks that corresponded to the turn switch points detected by the raters; consequently, we developed algorithms to find those particular landmarks.”

The results are for each signal in each cut-off filter option. From those results, we enhanced the first approach of the algorithm in order to develop an automatic one. We hope that with the new explanation in section 2.5 the procedure will be clearer and it will show that the algorithm, in this particular case, is a result. See L133-L150.

Detailed comment

Pag3 line 109-110, please clarify why this specific values of slopes were chosen (6.3°,5.7°)

A:  The inclination values were not based on literature. We wanted to check  the performance of the algorithm in different conditions so we used the  equipment available for us.

R:  You must give an explanation that is not “we used the condition  available in the equipment”, at least you must compare this value with  literature or infield data, and justify why you considered  this values sufficient to test the performance of the algorithm.

A: The values are smaller than the infield data. However, we wanted to add some variability to the data. The glider is not designed for use on an inclination, and cannot be further inclined without endangering the athlete. Therefore, we selected the maximum feasible inclination to ensure proper glider function and athlete safety, while introducing some degree of slope variability. In further steps of the algorithm development, the data will be recorded on the slopes and therefore the inclination will be “real”.

Reviewer 3 Report

The aim of this paper is to test various algorithms based on IMU data to define turn occurrence during skiing. The state of the art is correctly presented and the need to develop methods for which the sensors are not on the subject but on her(his) equipment is justified. The study is clearly and thoroughly justified.  Different combination of algorithms and filtering frequencies are tested.   

The structure of the article is surprising, with algorithms detailed after the results section.

The algorithms used to process acceleration are not detailed, which seems odd. It is important to point out the methods that do not work well to prevent others to reproduce them. Also, the presented results do not necessarily imply that no alternative algorithms based on acceleration could ever work.

The section describing the turn detection algorithms should be placed before the results.

Lien 148-160 The text is lengthy and hard to follow and the figure 2 explains things much better

Following the ANOVA, what type of post-hoc tests were conducted.

The authors stated that the left and average acceleration gave precision above one. This seems not to be the case for the average resultant acceleration filtered at 6 HZ which has the best measured precision (0.53). Please discuss.

Using only one subject to validate the algorithm is questionable. I advise to consider different subjects

Author Response

Reviewer #3:

The aim of this paper is to test various algorithms based on IMU data to define turn occurrence during skiing. The state of the art is correctly presented and the need to develop methods for which the sensors are not on the subject but on her(his) equipment is justified. The study is clearly and thoroughly justified.  Different combination of algorithms and filtering frequencies are tested.  

A: Dear reviewer, we would like to thank you for the valuable comments and hope to have made the necessary changes to improve this manuscript. The changes related to your comments are marked in “orange”.

The structure of the article is surprising, with algorithms detailed after the results section.

A: This is a concurrent topic for some reviewers, however, we believe that the algorithm is the main result of the study. However, we have modified the section 2.5 Data analysis in order to make the process clearer to the readers. We now wrote the main idea that we had while developing the algorithm in the methodology section, and we left the specific algorithm in the results section. We hope it reads better now. See L133-L150. Please also see answers to reviewers #1 & 2.

The algorithms used to process acceleration are not detailed, which seems odd. It is important to point out the methods that do not work well to prevent others to reproduce them. Also, the presented results do not necessarily imply that no alternative algorithms based on acceleration could ever work.

A: We completely agree on the importance of reporting what does not work as well. We have explained that not all algorithms were developed to the point of being automatic. As we answered in the previous question, the main idea used to detect the landmarks for all the signals is now explained in the section 2.5 Data analysis. We do not imply that an algorithm based on acceleration would never work, we state that with the landmarks selected, the precision of the turn switch points based on the acceleration signals were lower.

The section describing the turn detection algorithms should be placed before the results.

A: The algorithm is placed after the results table because it was developed following the conclusions extracted from that table. We hope that this explanation, combined with the previous ones (and the ones from the previous reviewers), makes clear our decision of placing the algorithm in the results section.

Lien 148-160 The text is lengthy and hard to follow and the figure 2 explains things much better

A: Thank you for your comment. That is precisely the reason why we felt the need of illustrating the measuring methodology.

Following the ANOVA, what type of post-hoc tests were conducted.

A: Bonferroni post-hoc tests were applied following the repeated measures ANOVA. See L126

The authors stated that the left and average acceleration gave precision above one. This seems not to be the case for the average resultant acceleration filtered at 6 HZ which has the best measured precision (0.53). Please discuss.

It is true that the average resultant acceleration filtered at 6 Hz has better precision than the average resultant gyroscope signal filtered at 3 Hz. However, when looking at the precision of the other filter options of the resultant acceleration, all the values are higher than >77.4 ms. Instead, the values related to the gyro resultant are all smaller than 14.5 ms. That is the reason why we discarded the acceleration resultant, we were very sceptical about the validity of that value. When visually inspecting the data, the acceleration signal was noisy; consequently, there were more peaks than turns, making the turn switch detection less reliable.

Using only one subject to validate the algorithm is questionable. I advise to consider different subjects

A: Please find below the answer written to reviewer #1: “The methodology proposed was with the aim of developing a first step towards an algorithm that works in the field. We wanted to determine which signal and filter cut-off options were more precise, and we believe the current methodology allow for those conclusions. The acquisition of more athletes in a laboratory set up, would definitely bring more variability to the data. However, this variability would be restricted by the plane of movement and the bindings of the glider. The variability that we would find on piste during on-snow skiing would be most likely based on different terrain conditions, techniques, vibration… and this is not measurable with the glider. We are planning the second part of this algorithm development which consists on evaluating and further developing the algorithm collecting data from different skiers in the field. This type of data collection will bring higher within and between subject variability to the signals and will allow us to enhance the algorithm with real skiing data. See L272-L280”

We hope this answers your comment.

Reviewer 4 Report

The paper is well structed and well written. The methodology is appropriate and well explained.

My only suggestion is to highlight the limitations of the study in the discussion, for example by discussing the possibility to find some issue when different participants will be tested.

Author Response

Reviewer #4:

The paper is well structed and well written. The methodology is appropriate and well explained.

My only suggestion is to highlight the limitations of the study in the discussion, for example by discussing the possibility to find some issue when different participants will be tested.

A: Thank you for your review and you encouraging words. We included your suggestion about the limitations in the discussion. We believe that while more subjects would bring more variability to the data, it would not be relatable to the variability found in field conditions where the plane of movement and technique are not limited. For this reason, we chose to perform the study with just one subject. See L272-L280

Reviewer 5 Report

This is a very nice paper, the proposed system is proving to be able to detect skiing turns. However, a further study could analyse these turns more in detail according to detect and compare the performance of the performer, e.g. an athlete compared to a moderate skiier. the same goes for the injury detetion, discussed in the introduction. However, this method is clearly exopsed, many relevant details are highlighted and the state of the art background is sufficient. Something else, in my sense, is missing: the numerical comparison of this detection method with the other existing patterns, which is diificult as regards to the main difference within the different frameworks. 

Line 107: small writing mistake, a space character is missing.

Author Response

Reviewer #5:

This is a very nice paper, the proposed system is proving to be able to detect skiing turns. However, a further study could analyse these turns more in detail according to detect and compare the performance of the performer, e.g. an athlete compared to a moderate skiier. the same goes for the injury detetion, discussed in the introduction. However, this method is clearly exopsed, many relevant details are highlighted and the state of the art background is sufficient. Something else, in my sense, is missing: the numerical comparison of this detection method with the other existing patterns, which is diificult as regards to the main difference within the different frameworks.

A: Thank you for your review and your recommendations. As you mention this turn detection methodology could be used to analyse and compare different conditions in further studies. Regarding to the comparison with other patterns, if I understand properly, you would like for us to compare the performance of this methodology with the previously reported ones. Unfortunately, the previous methodologies do not adjust their precision values (when reported) to the raters precision. We did compare our values to the previously reported ones without the correction.

Line 107: small writing mistake, a space character is missing.

A: Thank you, changed as suggested. See L107.

Round 2

Reviewer 1 Report

The paper has now improved of quality and is better. It could still be further improved and I'm looking forwards to your next paper where I strongly hope that you will address and solve the issues raised by the reviewers for this paper.